# Synchronization of gene expression across eukaryotic communities through chemical rhythms

Sara Pérez-García [1,2], Mario García-Navarrete [1,2], Diego Ruiz-Sanchis [1], Cristina Prieto-Navarro[1], Merisa Avdovic[1], Ornella Pucciariello [1] & Krzysztof Wabnik [1✉]

The synchronization is a recurring phenomenon in neuroscience, ecology, human sciences, and biology. However, controlling synchronization in complex eukaryotic consortia on extended spatial-temporal scales remains a major challenge. Here, to address this issue we construct a minimal synthetic system that directly converts chemical signals into a coherent gene expression synchronized among eukaryotic communities through rate-dependent hysteresis. Guided by chemical rhythms, isolated colonies of yeast *Saccharomyces cerevisiae* oscillate in near-perfect synchrony despite the absence of intercellular coupling or intrinsic oscillations. Increased speed of chemical rhythms and incorporation of feedback in the system architecture can tune synchronization and precision of the cell responses in a growing cell collectives. This synchronization mechanism remain robust under stress in the two-strain consortia composed of toxin-sensitive and toxin-producing strains. The sensitive cells can maintain the spatial-temporal synchronization for extended periods under the rhythmic toxin dosages produced by killer cells. Our study provides a simple molecular framework for generating global coordination of eukaryotic gene expression through dynamic environment.

[1] Centro de Biotecnologia y Genomica de Plantas (Universidad Politecnica de Madrid—Instituto Nacional de Investigacion y Tecnologia Agraria y Alimentaria), Pozuelo de Alarcon, Spain. [2] These authors contributed equally: Sara Pérez-García, Mario García-Navarrete. ✉email: k.wabnik@upm.es

Long-distance synchronous behavior exists in biological systems such as brain circuits[1–3], ecosystems[4,5], human networks[6,7], developmental clocks[8–10], and circadian rhythms[11,12]. Typically, synchronization in biological systems may result from a coupling between individuals and can effectively compensate the variation in molecular components and underlying biochemical processes[13,14]. For instance, synchronized responses can emerge from local interactions between surface receptors that assemble molecular bridges between adjacent cells[15,16] or a directional movement of small molecules between cells and tissues capable of coordinating gene expression during developmental progression[17]. Therefore, designing efficient synchronization strategies bears a key to control global dynamics in complex multicellular environments, such as microbiomes, bioreactors, organoids, and tissues.

Over the last decade, substantial progress has been made in the emerging field of synthetic biology, leading to the development of effective strategies for synchronization of bacterial consortia through quorum sensing[18,19], error-reducing design of gene circuits[20], gaseous exchange[21], self-organizing local interactions[22], signal amplification[23], or entrainment[24]. Most of these engineered systems rely on the intrinsic oscillatory dynamics of synthetic gene circuits that were tailored for bacterial cells, imposing further restrictions for the effective implementation beyond prokaryotes. Consequently, robust strategies for long-distance synchronization in eukaryotes remain largely unexplored.

It is well established that many bacterial species deal efficiently with antibiotic-related stresses by coordinating detoxification through stress-responding genes from multi antibiotic resistance (Mar) family[25,26]. We thought that this bacterial strategy[25,26] for antibiotic stresse response could be adapted to eukaryotic hosts to control coordination of gene expression among cell populations. Therefore our aim was to construct a chemically controlled system that would sustain the long-term spatial–temporal synchronization of gene expression among eukaryotic cells under diverse environmental conditions. To further foster portability concerning a broad spectrum of hosts, an ideal design should be simple enough and avoid additional complexities such as intercellular coupling or complicated regulatory architectures.

Here, we implement the synthetic synchronization mechanism in the model eukaryote yeast S. cerevisiae that is based solely on two Mar receptors for small molecules indole-3-acetic acid (IAA, auxin) and salicylic acid (SA)[25] and a fluorescent reporter. The comprehensive analysis of the spatial–temporal dynamics of a reporter gene under IAA and SA applications reveals the unexpected mechanism underlying receptor activity that provides the robust coordination of gene expression among yeast communities that depends on the rate-dependent hysteretic switching. Theory and experiments jointly demonstrate the efficient strategy for inter-population control mediated by chemical rhythms in a dynamically changing environment.

## Results

**Bacterial receptor responses are synchronized through chemical rhythms in yeast.** Our synthetic system is composed of a pair of multi antibiotic resistance (Mar) receptors for SA[25] and IAA[27,28] (Fig. 1a). These compact proteins contain DNA binding and ligand-responsive dimerization domains (Supplementary Fig. 1). Briefly, MarR or IacR receptors (Fig. 1a) were fused with a strong herpes simplex virus trans-activation domain (VP64)[29] to produce transcriptional activators in yeast. A degradation tag[30] was placed at the carboxy-terminal of both receptors to increase their turnover rates and thereby a dynamic range of the whole system (Fig. 1a and Supplementary Fig. 2a). To record the output

from our system we used a short-lived green fluorescent protein reporter (dEGFP)[31] controlled from a minimal promoter that integrates MarR and IacR operator sequences[25,28] upstream of TATA-box (Fig. 1a).

We found that MarR and IacR showed specific responses to either SA or IAA in yeast cells with a dynamic range of 2-orders of magnitude (Fig. 1b, c); $EC_{50} = 0.13\,\mu M$ (IAA) and $407.3\,\mu M$ (SA). IacR response was specific to IAA and insensitive to SA (Supplementary Fig. 2c). On contrary, MarR responded to SA but did not respond to IAA (Supplementary Fig. 2d), confirming the orthogonality of both receptors.

To further explore spatial–temporal dynamics of MarR and IacR activity in yeast populations we performed system analysis under controlled environment generated in a microfluidic device (Supplementary Fig. 3a–c) (see Supplementary Information for details). We recorded a transient dynamics of dEGFP reporter in response to the rectangular pulse (Fig. 1d, e, gray box) of SA and IAA that revealed bursts of dEGFP signal observed across yeast cell populations, demonstrating a time lag in response to the applied signal (Fig. 1d, e). Next, we quantified this effect using precision metrics that was defined as the relative differences between the duration of dEGFP burst and the duration of SA or IAA rectangular pulse (Fig. 1f). Clearly, dEGFP reporter deactivation mediated by chemicals exhibited a time lag in the dEGFP signal decay with respect to chemical input which was further confirmed by the reduced response precision (<80%) (Fig. 1f).

Next, we checked whether we could capture the transient switching of dEGFP reporter by sequential application of both chemicals. For that, we build a gene circuit that allowed the antithetic control of dEGFP by SA and IAA by converting MarR into transcriptional repressor (Fig. 1g and Supplementary Fig. 2b). Initially, we confirmed using microwell plate fluorescence scanning assays and microscopy that the robust gated switching was present using combinatorial chemical stimuli (Supplementary Fig. 4a, b). We then studied system dynamics under out-of-phase pulses of SA and IAA. The dEGFP reporter dynamics revealed a rapid activation and slow decay (Fig. 1h, i), further confirming the dependence of reporter output on the history of chemical input; future state of the system depends on its past (Fig. 1h) that is key characteristic of hysteretic systems[32–36]. In engineering field such hysteretic mechanism is employed to effectively filter noise and thus tune output precision[37]. In addition, regardless of inherent noise in receptor abundance and biochemical reactions we observed the synchronized dEGFP dynamics across spatially distant yeast colonies (Fig. 1i).

**MarR and IacR may undergo conformational switching.** Employing the Mar-based strategy inspired by coordinated stress response in bacteria allowed the synchronous switching of dEGFP reporter expression in yeast populations (Fig. 1h, i). However, this system does not incorporate positive feedback that could potentially produce a time-lag and history dependence observed in our experiments (Fig. 1h). To identify a potential origin of metastability associated with a putative hysteretic mechanism, we performed all-atom molecular dynamics (MD) simulations in three independent replicates using crystal structures of MarR proteins (RCSB codes: 3VOE [https://doi.org/10.2210/pdb3VOE/pdb], 1JGS [https://doi.org/10.2210/pdb1JGS/pdb], 5H3R [https://doi.org/10.2210/pdb5H3R/pdb]). Remarkably, MD simulations predicted that MarR apoprotein structure remains largely flexible and fluctuates without showing clear marks of a rigid conformation (Fig. 2a, d, Supplementary Movie 1). Thus, apoprotein can adapt to the shape of the DNA operator (ON state) by successively adjusting a distance between DNA binding (DB) domains (Fig. 2b, d, Supplementary Movie 1). In contrast, the

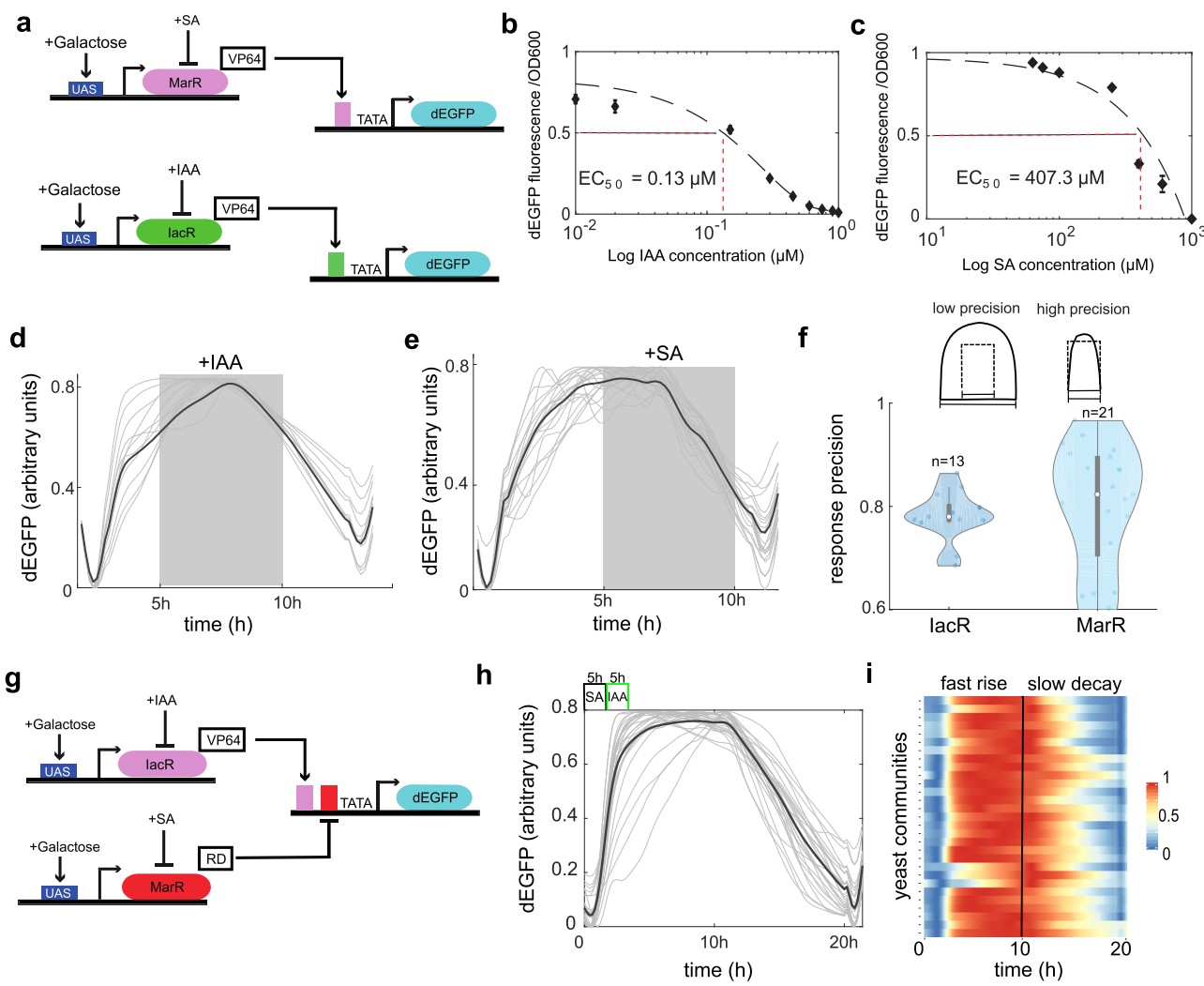

**Fig. 1 Mar-type receptors show the specific nonlinear switch-like dynamics in the presence of SA and IAA pulses. a** Schematics of SA (top) and IAA (bottom) receptors. MarR (pink) or IacR (green) repressors are tagged with a synthetic VP64 viral transactivation domain, nuclear localization signal (NLS), and PEST degron. Galactose is used to induce the transcription of both receptors through galactose-responsive UAS sequences(blue). The green fluorescent reporter (dEGFP, cyan) contains an uncleavable ubiquitin (G76V) tagged to N-terminus of enhanced GFP (EGFP) that is driven by the minimal promoter with either MarR (pink bar) or IacR (green bar) operators placed directly upstream of TATA-box. **b**, **c** Response kinetics of IacR module to IAA (**b**) and MarR module to SA (**c**) are shown. Note a dynamic range of two-orders of magnitude for physiological concentrations of SA and IAA. Means (diamonds) and standard errors bars are from three independent replicates. **d**, **e** dEGFP fluorescence time profiles for IacR (**d**) and MarR (**e**) receptors. Stimulus was provided as indicated: 5 h (gray box) chemical pulse followed by 5 h reset phase (10 h cycle). Note, time-lag in both activation and repression of dEGFP reporter. **f** Response precision plots for both receptors ($n = 13$, $n = 21$ traps). In the scheme, the input pulse is shown with a dashed rectangular profile and expected dEGFP output response with a solid curve. Violin plots represent medians (white dots), interquartile zones (gray bars) and 95% confidence levels (solid gray line). Gray contours show density plot boundaries. **g** Schematics of dual-receptor circuit with IacR activator (pink) and MarR repressor (red) used to explore switching dynamics. **h** dEGFP dynamics are controlled by successive switching of SA and IAA levels (amplitudes: 500 µM SA and 1 µM IAA) in the 10-h period. Mean profiles across yeast colonies ($n = 29$) are shown as black line. Note a characteristic switch-like response dynamics. **i** A heat map (0–1) of normalized dEGFP fluorescence in distant communities ($n = 29$). Synchronous switching was observed across yeast colonies. For refence dashed line indicates the end of chemical stimulus cycle. Source data are provided in the Source Data file.

presence of SA in the MarR dimerization interface brought monomers close to each other, creating a rigid closed conformation (Fig. 2c, d, Supplementary Movie 1) that was unable to bind DNA (OFF state). We calculated the probability distribution of the distance between R73 residues of DB domains that are in direct contact with DNA. Apoprotein was able to scan across various distances and eventually adapted the ON state (Fig. 2e).

To address further the dynamics of conformation changes, we performed the metadynamics energy landscape analysis with the forced local perturbation of distance between R73 atoms of both monomers using Morse potentials. Time-dependent changes of energy portraits in apoprotein and DNA-bound MarR

configuration (ON state) revealed the presence of a stable energy minimum (M), which was absent in holoprotein simulations (Fig. 2f). This minimum corresponds to the smallest deviation from the reference distance of ≈20 Å between DB domains, as observed in crystals of DNA-bound MarR (Fig. 2e).

Next, we asked if IacR could respond to IAA using a analogous conformational switch mechanism. The best-scoring homology model of IacR based on MarR-derived templates (Supplementary Fig. 5) supported by all-atom molecular docking predictions of putative IAA binding pocket was in a fair agreement with crystal structures of other Mar-type receptors (Supplementary Fig. 6a, b). Model predictions suggest that apoprotein may remain flexible

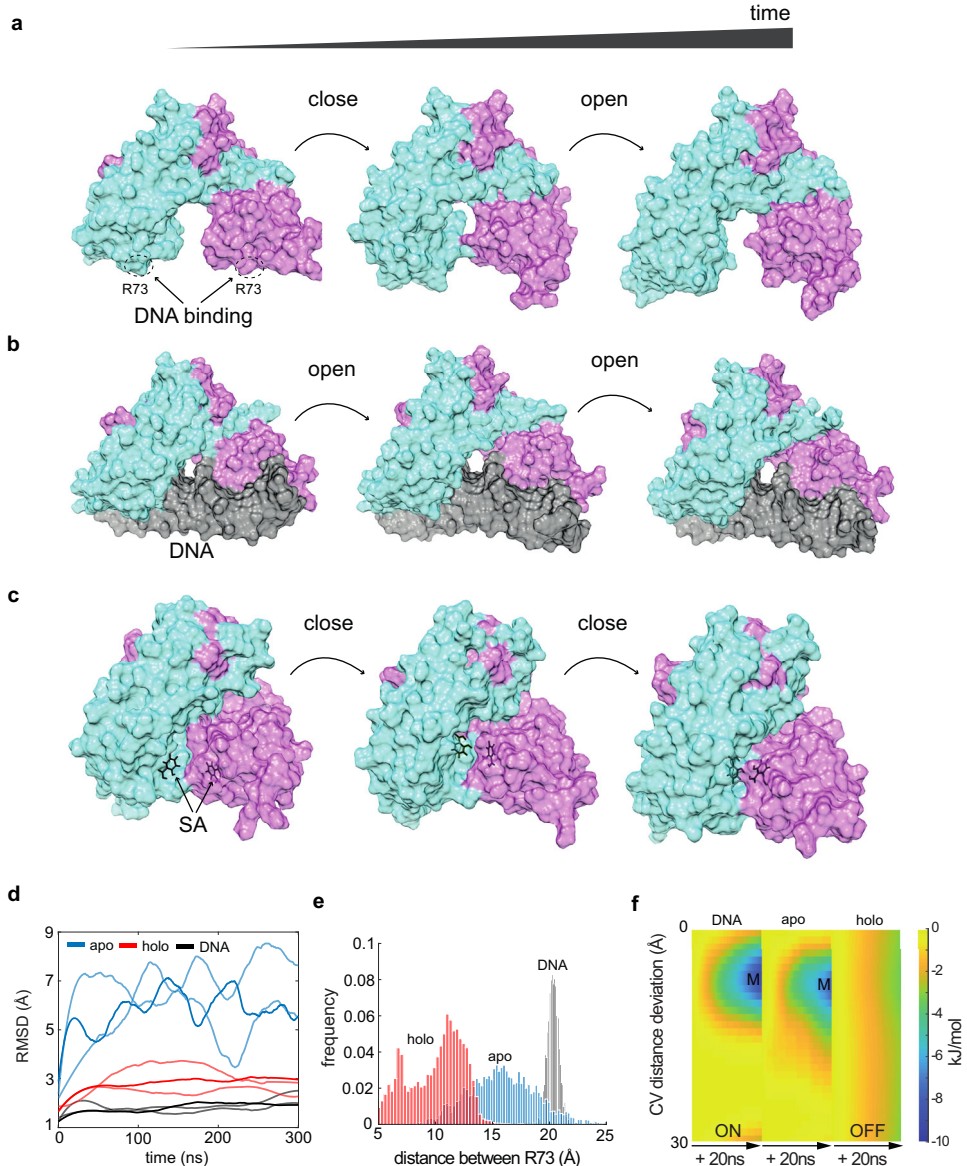

**Fig. 2 On–OFF states underlies the chemical sensing by Mar receptors. a–c** Time-lapse from MD simulations of the flexible MarR apoprotein (monomers are shown in cyan and purple) (**a**) display a cyclic change of conformation states (open–closed–open). MarR dimer bound to DNA operator site (black) remains in the open conformation with a specific distance of ~20 Å between DNA contacting R73 (**b**). MarR bound to two molecules of SA(black) (**c**) display successive shortening of the distance between R73 that reflects an unbinding of MarR from DNA. **d** Time-traces of molecular dynamics simulations with MarR$_{apo}$ (blue), MarR$_{holo}$ (red), and DNA-bound MarR (black) (3 independent replicates are shown per modeled system). The root-mean-square deviation of atomic positions (RMSD) is shown for dimer simulations of 300 ns. Note, the cyclic pattern of closed-to-open conformation of MarR$_{apo}$ which was absent from MarR$_{holo}$ simulations as holoprotein stabilizes in the closed conformation. In contrast, DNA-bound MarR retains an open confirmation. **e** A probability distribution of the mean distance between R73 atoms of both monomers (from all three replicates) for apo (blue), holo(red), and DNA-bound (black) configurations. Note that the holo state provides $a = 2$-fold short distance between DBD domains than that required for a stable bond with DNA. **f** Energy landscapes for three MarR configurations (apo, holo, DNA-bound) for deviation from the preferred distance of ≈20 Å between R73 (both chains) required for DNA binding. Note energy minima wells (M, blue) for apo and DNA bound configurations which were absent in holoprotein simulations indicating that the ligand-bound MarR did not reach an energy minimum required for the association with the DNA configurations.

concerning the spacing between DB domains (Supplementary Fig. 7a). In contrast, the predicted IAA binding pose (Supplementary Fig. 6a) indicates the extension of a distance between these domains (Supplementary Fig. 7a–c), as demonstrated by the frequency of distance distribution (Supplementary Fig. 7d) followed by metadynamics analysis (Supplementary Fig. 7e). Therefore, these theoretical predictions of IacR dynamics indicate common design principles of DNA recognition with MarR, including backbone flexibility and successive switches of

conformation three states in the presence of SA and IAA. Collectively, our model predicts that MarR and IacR receptors can switch between two stable and one flexible states that is dependent on chemical fluctuations. This putative metastability at the structure layer could potentially explain time-lagged hysteretic response observed in our experiments. Similar structural hysteresis concept has been demonstrated in voltage-gated ion channels[38–40] which can facilitate the long-distance electrochemical synchronization within[41] and between bacterial biofilms[42].

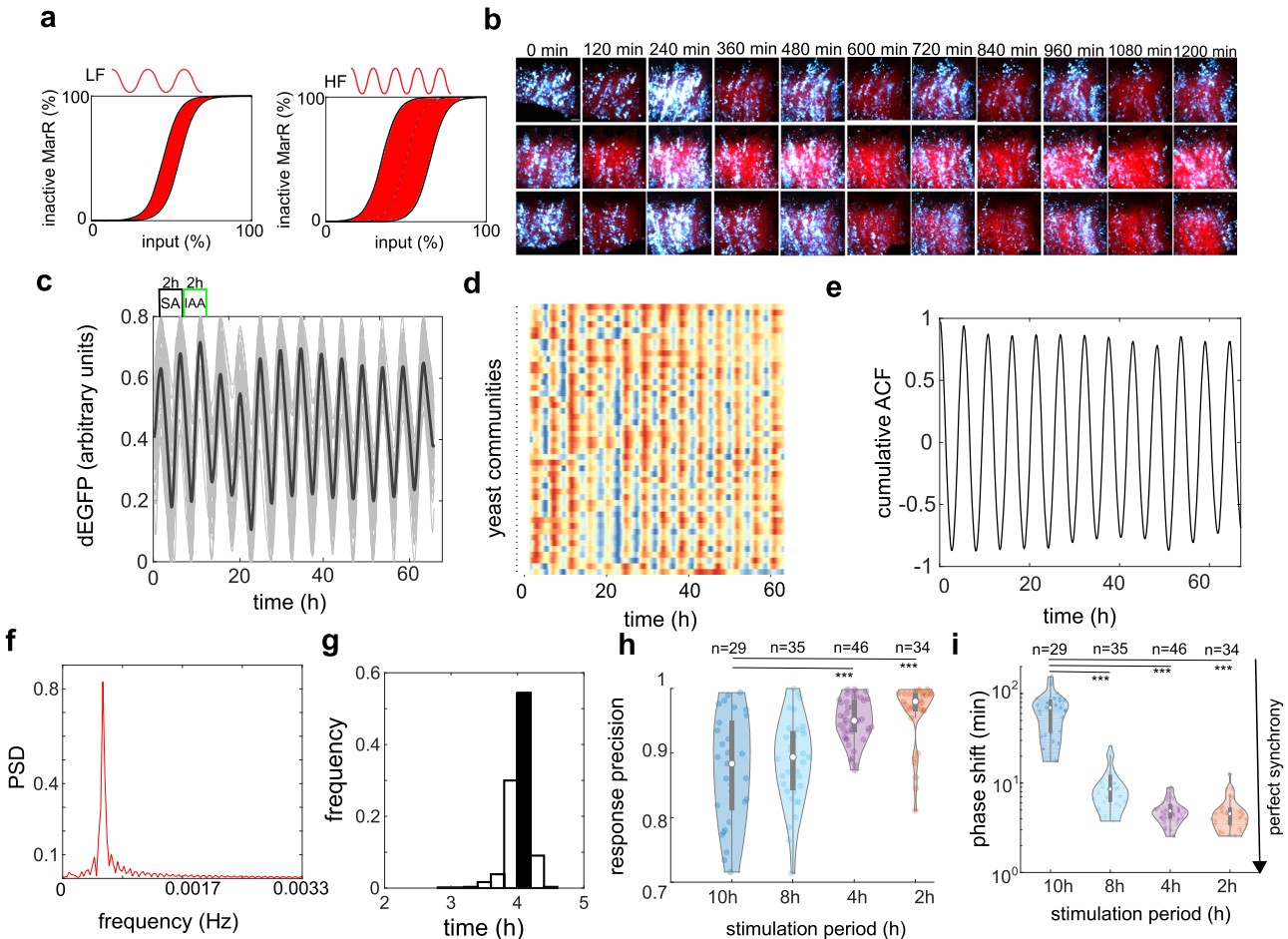

**Fig. 3 Synchronous dynamics and response precision across spatially isolated yeast communities. a** A theoretical model of rate-dependent hysteresis guiding SA and IAA perception dynamics. Generally, high frequency (rate) of input signal leads to an enlarged hysteresis (red region), reducing noise impact on the output switching and thus improving the synchrony. **b–d** Experimental validation of theory (**a**). **b** Example image snapshots from microfluidic experiments with 4 h stimulation cycles (each row represents time evolution of individual yeast community). dEGFP is shown in cyan and exponentially growing cells in red. Note a full synchrony between colonies (**c**), time evolutions of reporter fluorescence in spatially distant yeast communities. Mean dEGFP trace for all communities is shown in black, and corresponding heat maps (**d**) for all recorded communities ($n = 46$). Note, a remarkable synchrony and negligible phase drift between communities. **e** Cumulative autocorrelation function (ACF) calculated from all recorded communities ($n = 46$) shows the robust synchronous pattern at the inter-colony level. **f** Power spectra density (PSD) analysis (related to **e**) indicates a convergent period of response (black bar) in different yeast communities. **g** Probability distribution of periods for input stimulus in cycles of every 4 h (**h**, **i**). Violin plots of Response precision (**h**) (*** p-values $= 3.2^{-7}$, and $2.0^{-6}$, a one-way ANOVA with post-hoc Tukey's HSD), and phase drift (**i**) as a function of increasing frequency of stimulus (*** p-values $= 1.1^{-10}$, a one-way ANOVA with post-hoc Tukey's HSD). The description of violin plots is as in Fig. 1f. A heat map legend is as in Fig. 1i. Source data are provided in the Source Data file.

**Robust and tunable synchrony across yeast communities.**
Time-lapse microscopy experiments and all-atom computer simulations together support the notion of putative hysteretic mechanism in MarR- and IacR-mediated activation of reporter expression. Next, we asked whether this mechanism involves persistent memory (classical hysteresis) or alternatively time-lag changes with the frequency (rate) of SA and IAA applications (transient rate-dependent hysteresis)[43] (Fig. 3a). To test this scenario, we performed independent microfluidics experiments by gradually decreasing stimulation period from 10 to 2 h. Yeast cell populations (Fig. 3b) oscillated in the near-perfect synchrony on millimeter scales following gradually faster chemical rhythms (Fig. 3c, d, Supplementary Movie 2). We reported successive cycles of rate-dependent time-lagged switching (Supplementary Fig. 8a, b) by measuring a time lag between chemical input and dEGFP reporter expression in microfluidics experiments. Indeed, we found that a hysteresis was enlarged with the higher speed of chemical stimuli (Supplementary Fig. 8c). Next,

we complemented these observations by quantifying three distinct metrics across yeast communities for different stimulation periods (Supplementary Fig. 9a): cumulative (global) auto-correlation function (Fig. 3e, and Supplementary Fig. 9b), global frequency spectra (Fig. 3f, and Supplementary Fig. 9c) and probability density distribution of periods (Fig. 3g and Supplementary Fig. 9d). The precision of response was significantly improved with the higher frequency of applied input (Fig. 3h) accompanied by sharp period distribution (Supplementary Fig. 9d) but mean amplitude reduction (Supplementary Fig. 9e). Quantifications of phase drift between communities confirmed a remarkable improvement of synchronization; from ~20% phase drift (10 h) to only 2% phase drift (2 h and 4 h) (Fig. 3i), thereby further supporting the rate-dependent switching mechanism (Fig. 3a). This robust synchronization did not depend on the particular choice of expression system (i.e., multicopy episomal or single-copy genomic integration) (Supplementary Fig. 10a–e). Moreover, fixing both chemicals (Supplementary Fig. 11a) or

either of them (Supplementary Fig. 11b, c) at predefined concentration had detrimental effects on the response precision, and more eminently on the synchronization (Supplementary Fig. 11d, e). These findings indicate that both components of the circuit synergistically contribute to the robust synchronization dynamics observed in our experiments.

Our findings indicate that hysteresis is not permanent but instead it changes with the rate of chemical inputs. An underlying mechanism by which the hysteresis region expands with the increased frequency rate of input (Fig. 3a) refers to the rate-dependent hysteresis in engineering and it is used for precision control of piezoelectric actuator systems[43]. To understand benefits of such mechanism for noise gating and better precision in the coordination of cell responses we developed a simple model of stochastic switching with hysteresis based on the theory of stochastic resonance in noisy bistable systems[44–46]. We show that if the hysteresis region of noisy bistable systems increases with input frequency rate, a robust synchronization can be achieved for a wide range of noise levels (Supplementary Fig. 12a). Theory also suggests that weaker driving will deteriorate synchronization (Supplementary Fig. 12b) which we then confirmed experimentally by reducing amplitude of chemical stimulation by a 10-fold (Supplementary Fig. 13a–f). Therefore, theory and experiments jointly indicate that a rate-dependent hysteresis encoded in our system may control the level of synchronization among yeast populations.

Next, we then asked whether amplitude, precision, and synchronization could be additionally tuned by an incorporation of feedback (closed loop) in our synthetic system (Fig. 4a). In particular, we tested two scenarios by introducing negative feedback (NFL) (Fig. 4b) or positive feedback (PFL) (Fig. 4c) in our synthetic circuit design. Initially, we performed well-plate fluorescence measurements and found that overall population response was generally faster (as expected from addition of feedback) in either of closed loop circuits compared to the open-loop system (Supplementary Fig. 14a–c). By time-lapse analysis of three circuit variants in microfluidic device showed that synchrony can be further tuned by implementation of closed loop architecture (Fig. 4a–c, Supplementary Movie 3). In particular, quantifications of dynamics in closed-loop circuits revealed a tunability of NFL circuits with respect to response amplitude (Fig. 4d), precision (Fig. 4e), and synchrony level (Fig. 4f).

Therefore, closed loop regulations can increase the speed and fine tune response characteristics without compromising inter-population level synchronization.

**Inter-population synchrony is maintained under pulsatile stress.** Engineered synthetic systems for the coordinated gene expression in eukaryotic consortia should ideally provide opportunities to control the adaptation to dynamic environmental challenges such as the presence of toxins and other harmful agents. Previously established synchronization strategies has not been yet tested under stressful environments.

To test whether our synchronization strategy would maintain coordinated gene expression in the presence of a dynamic stress, we constructed a synthetic dual-strain ecosystem responsive to IAA and SA inputs (Fig. 5a). 'Sensitive strain' carries an unmodified gene circuit as presented in Fig. 1g whereas the yeast 'killer strain' produces viral K1 toxin[47] exchanging the reporter gene (Fig. 5a). For reference we also constructed a disarmed strain (empty) that carries empty plasmids that we simultaneously co-cultured with the sensitive strain (Supplementary Fig. 15a). Co-culturing sensitive and killer strains in well-plates (Supplementary Fig. 15b) and in the microfluidic environment revealed a

complex interaction patterns within this synthetic ecosystem (Fig. 5b, Supplementary Movie 4). Our data indicate that both communities could co-exist for an extended time, simultaneously in distant regions of the chip under periodic stimulations with chemicals (Fig. 5b, Supplementary Movie 4). While the 'killer' strain dynamics mimics that of sensitive strain cultured alone (Fig. 4a) the frequency response of sensitive strain showed altered pattern (Fig. 5c, d) compared to the control experiment (Supplementary Fig. 15a). In particular, we observed a successive period doubling events during more than two days of ecosystem culturing in the microfluidics chip (Fig. 5c) and was absent from the control co-culture with the empty strain (Supplementary Fig. 15a). Global autocorrelation function (Fig. 5e) and power spectra analysis (Fig. 5f) demonstrated a clear shift of response period towards lower frequencies only in cocultures with killer strain (Fig. 5f), again indicative of successive period doublings. Quantifications of the ratio between dEGFP and SA/IAA signal periods which further confirmed series of period doublings (Fig. 5g). However, despite of observed period doublings, we found that the synchronous response was prevalent among sensitive communities (Fig. 5d and Supplementary Fig. 15c) but with reduced precision as compared to control experiments (Fig. 5h).

These findings indicate that sensitive cells may respond with the series of period doublings to toxin presence but at the global scale, sensitive populations are capable of temporally maintain coordinated gene expression, presumably by increasing chances of survival. Similar measures may be taken by bacteria when dealing with antibiotic stresses[25,26], thereby demonstrating a design principle underlying our synchronization strategy.

## Discussion

We present a robust mechanism for the chemical coordination of spatially isolated eukaryotic communities that does not rely on intercellular coupling. Despite its simplicity our system displayed a remarkable synchronization level (~98% of communities were synchronized) in growing populations under the dynamic environment. Synchronization strategies previously established in bacteria can produce good synchrony levels but required strong long-distance coupling to achieve the robust inter-population synchrony[18–23]. It is also unclear if these strategies can be transplanted in more complex organisms such as eukaryotes, as well as if they would remain robust in stressful environments.

Due to structurally encoded rate-dependent hysteresis, our system can be tuned with the rate of chemical stimuli to deliver precise synchronization of eukaryotic populations. Similar concept applies to piezoelectric actuators which can convert electrical signals into precise movement without requirement for the permanent memory. Also, the proposed mechanism differs from noise-induced synchronization also known as stochastic resonance because the increased frequency of input attenuates the synchronization in classical bistable systems[44,45].

Although demonstrated in yeast, our strategy may offer a potential for the future implementation in other eukaryotes such as animals or plants because it requires only MarR, IacR, and small chemical molecules to control the coordination of gene expression in eukaryotic cell communities. Both receptors can be tagged on the c-terminus with compact activation or repression domains (i.e., Krüppel/KRAB or EAR) or chromatin regulators such as active units of histone acetylases and deacetylases derived from animals or plants, thereby extending the spectrum of transcriptional regulation this system has to offer. Furthermore, these two receptors could be implemented in plants as a direct biosensors for readout of major plant hormones auxin and salicylic acid.

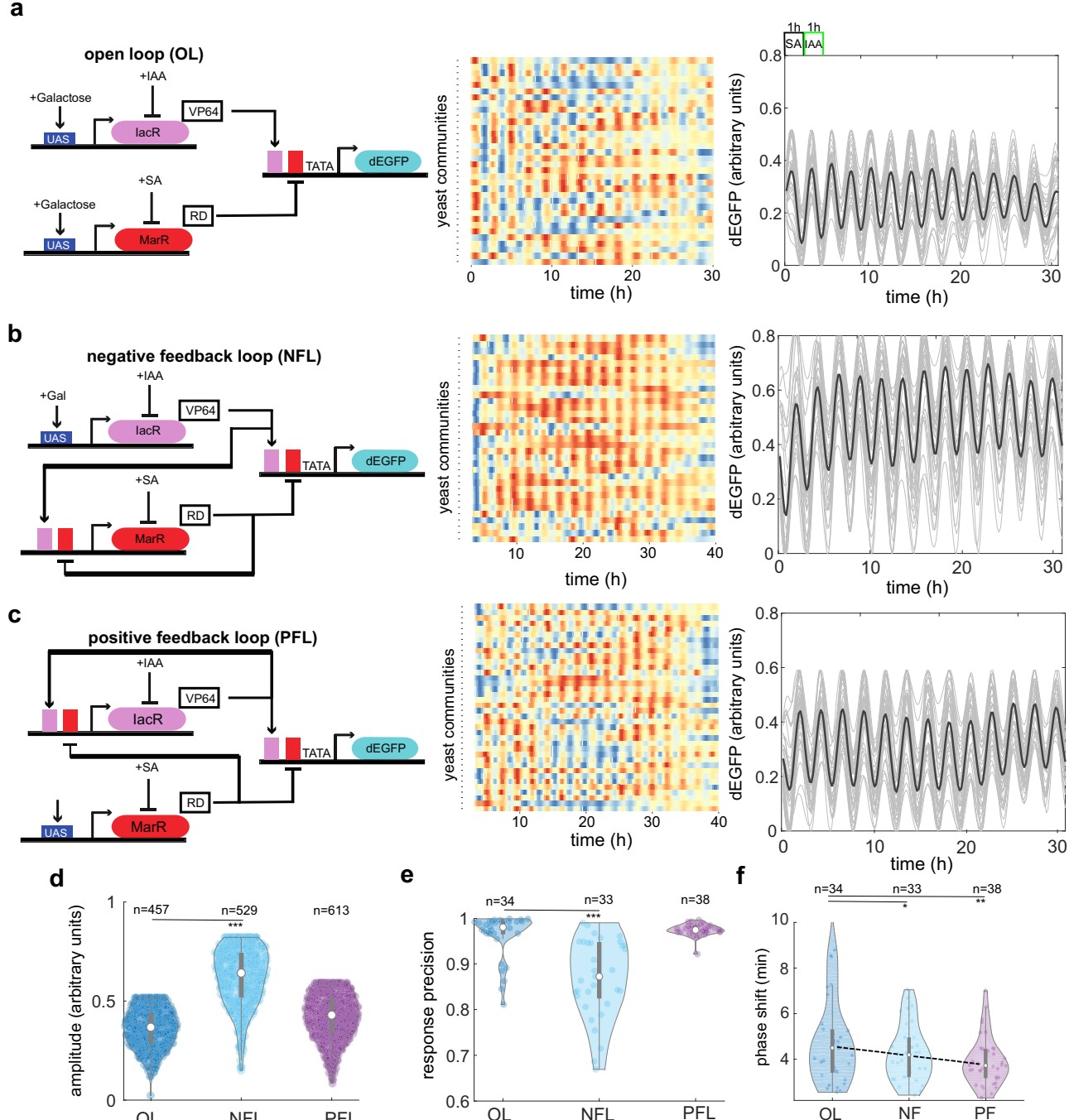

**Fig. 4 Closed-loop circuits offer an extended tunability of response in synchronized cell populations. a–c** Circuit schematics related to Fig. 1g (left panel), synchronous dynamics across yeast communities visualized with heatmaps of dGFP fluorescence over time (middle panel), and time-lapse profiles with mean trends (right panel black lines) are shown for open-loop system (OL) (**a**) and two closed loop system with either negative feedback (NFL) (**b**) or positive feedback (PFL) (**c**). Color coding scheme is as in Figs. 1 and 3. **d** Probability distribution of amplitudes for three different circuit variants. Note an ~50% increase of amplitude in NFL systems as compared to the OL system (***$p$-value $= 1.1^{-10}$, a one-way ANOVA with post-hoc Tukey's HSD). **e** Amplitude increase in the NFL system is counteracted by reduced response precision (***$p$-value $= 9.4^{-9}$, a one-way ANOVA with post-hoc Tukey's HSD). **f** Inter-community synchrony is further improved by including feedback loops in the original circuit (*$p$-value $= 9.0^{-2}$, **$p$-value $= 9.0^{-3}$, a one-way ANOVA with post-hoc Tukey's HSD). The description of violin plots is as in Fig. 1f. A heat map legend is as in Fig. 1i. Source data are provided in the Source Data file.

We tested our strategy against environmental odds in the synthetic ecosystem that combined chemical and toxic rhythms. We showed that a robust synchronization among communities prevails in stress bearing environments while response frequency can be modulated by periodic toxin dosage produced by the killer yeast. Thus, our system is capable of guiding a collective synchrony and temporal stabilization under these particularly harmful environmental conditions. Bacteria may use a similar strategy to fight against antibiotic threats. Moreover, by exchanging toxins with therapeutic agents our system could serve a foundation for designing a tool for synchronized release of drugs and control the population of cancer cells among other therapeutically important applications.

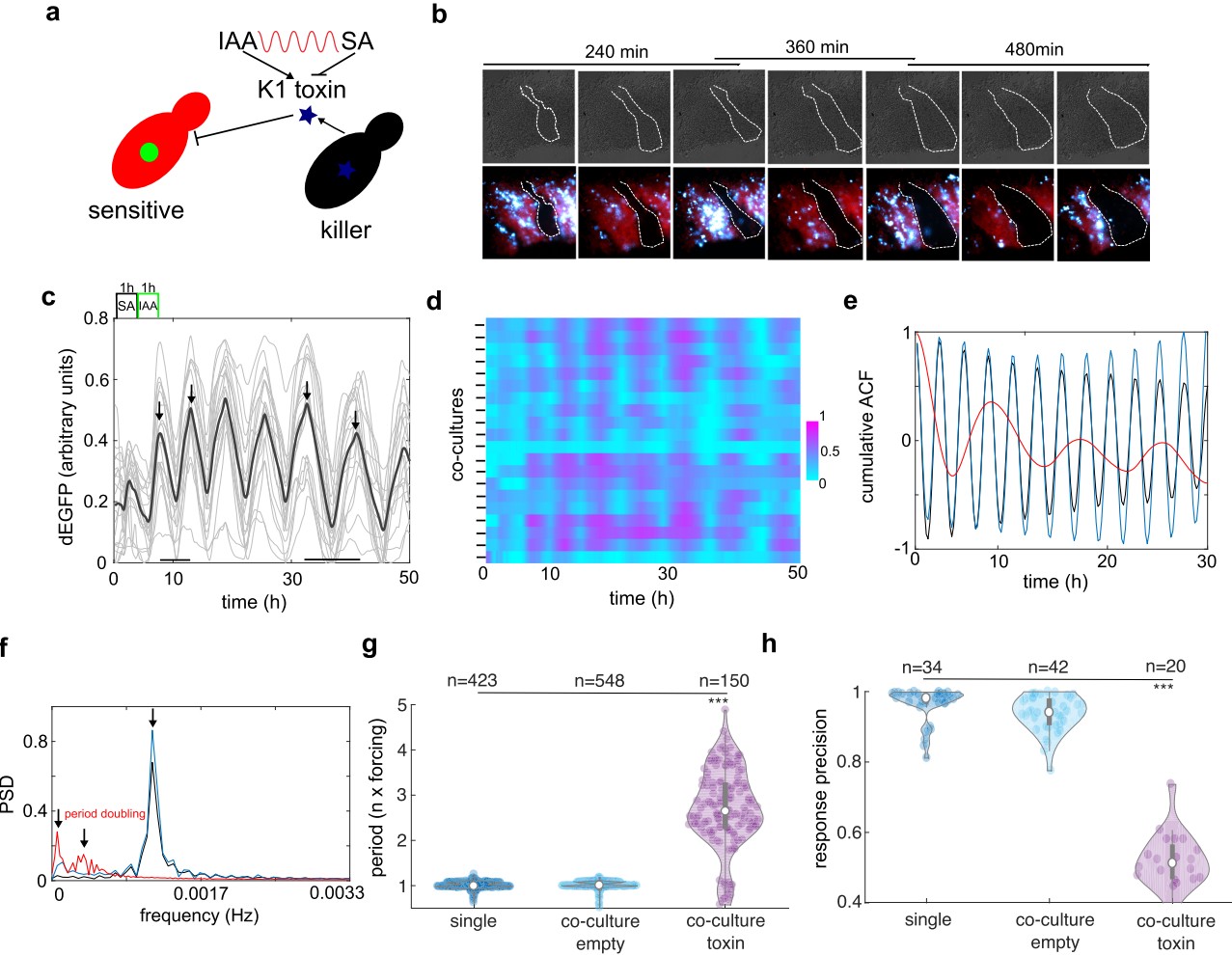

**Fig. 5 Synthetic yeast consortium maintains synchrony under chaotic regime in the presence of pulsatile toxin stress. a** Schematics of designed synthetic yeast ecosystem driven by SA-IAA chemical clock. Two sister strains one producing green reporter (sensitive, red) and other producing K1 toxin (blue star) coexists under rhythms of IAA and SA. **b** Snapshots from time-lapse microfluidics experiments with co-cultured sensitive and killer strains under 2h-lasting input pulses. Note successive period doublings in three successive cycles. Dark regions with dashed borders represent killer strain adjacent to sensitive counterparts (red and cyan). **c**, **d** Time-lapse profiles of dEGFP with mean trends (right panel black lines) for sensitive strain (**c**) and corresponding cold heat map (cyan-purple) with normalized reporter between 0 and 1 (**d**). Note, period doublings (arrows) and simultaneous prevalence of synchrony between recorded communities in the microfluidic chip ($n = 20$). **e** Cumulative autocorrelation function for sensitive strain grown alone (black line, reference), co-cultured with strain carrying empty plasmids (blue line) and co-cultured with killer toxin-producing strain (red line), note the absence of period doubling in toxin-free ecosystems. **f** Power spectra density demonstrate left shifts towards lower frequencies (period doublings) in toxin-producing co-cultures (red line) as compared to controls (black and blue lines). **g** Distribution of period doublings in the sensitive strain grown alone, or with coculture with and without toxin producing sister strain. (*** $p$-value $= 1.1^{-10}$, a one-way ANOVA with post-hoc Tukey's HSD). **h** Response precision drops by nearly 50% in the co-culture experiments with toxin producing strains but not in the controls (*** $p$-value $= 1.1^{-10}$, a one-way ANOVA with post-hoc Tukey's HSD). The description of violin plots is as in Fig. 1f. Source data are provided in the Source Data file.

This work also provides a mechanistic basis for de novo engineering synchronous dynamics in complex living communities, such as microbiomes for various synthetic biology applications, including control of coordinated production in the industrial bioreactors and production and release of therapeutics and vaccines. Lastly, this study provides a mechanistic principle for designing synthetic gene circuits for synchronization in higher organisms such as plants and animals, thereby expanding the repertoire of possibilities explored through evolution.

## Methods

**Structural modeling and molecular dynamics simulations.** Crystal structures of the apoprotein, DNA-bound and holoprotein were obtained from Protein Data Bank RCSB database (RCSB codes: 3VOE [https://doi.org/10.2210/pdb3VOE/pdb], 1JGS [https://doi.org/10.2210/pdb1JGS/pdb], 5H3R [https://doi.org/10.2210/pdb5H3R/pdb]). In preparation for MD simulations, structures were processed using Charmm-gui (www.charmm-gui.org) Solution Builder. Simulation-ready structures contain hydrogens, water box around the structure fitted to the size of system, and water environment with 0.15 M NaCl ions and an isothermal-isobaric (NPT) ensemble with a constant amount of substance ($N$), pressure ($P$), and temperature (T). The pressure was set at 1 atm (1.01325 bars) and temperature at 303 K. MD simulations were performed using OpenMM v7.4 environment (http://openmm.org/) with hydrogen mass repartitioning (HMR) scheme. Initial steps of minimalization and equilibration for 1 ns were followed by tracking 300 ns conformation dynamics with the time step of 2 femtoseconds (three independent replicates per system). OpenMM protocol uses default the cutoff used for treating the non-bonded interactions of 12 Å. Particle Mesh Ewald 2 s (PME) periodic boundary conditions were used for the electrostatic force calculations. Trajectory files from the simulations were processed with VMD v1.9.3 (www.ks.uiuc.edu/Research/vmd/) to generate RMSD plots and atom distance distributions. The visualization of structures was done in UCSF Chimera v1.3 (https://www.cgl.ucsf.edu/chimera/). Metadynamics energy landscape analysis was performed in OpenMM using Metadynamics module and Morse/long-distance potential between atomic centroids of R73 (MarR) and K80 (IacR) residues in DB domains. This collective variable (CV) is calculated based on the reference distance of ≈20 Å

inferred from the crystal structure of DNA-bound MarR as an equilibrium parameter in the Morse potential function. 300 ns trajectories in triplicates were used as a starting point to perform an additional 20 ns simulation (total 320 ns) with CV in the flexible range of 0–3 nm.

The IacR structure models were obtained with Swiss-Model using several templates, RCSB codes: 3VOE [https://doi.org/10.2210/pdb3VOE/pdb], 1JGS [https://doi.org/10.2210/pdb1JGS/pdb], 5H3R [https://doi.org/10.2210/pdb5H3R/pdb]. Analysis of IacR model confidence against experimentally resolved crystal structures as well as structural alignment with confidence levels in regions of interest (i.e., DNA binding, dimerization) are shown and discussed in Supplementary Fig. 5.

**Structure alignment and quality of structure prediction**. Molecular Evolutionary Genetics Analysis (MEGA) 11 (https://www.megasoftware.net/) was used to produce the alignment of the multiple MarR proteins, specifically the ClustalW algorithm available in MEGA. The Gap Opening Penalty was set at 10.00 and the Gap Extension Penalty at 0.20. The protein weight Matrix used in this multiple alignments was Gonnet. For the visualization and generation of the alignment image, we used the Jalview program with the input alignment file from MEGA.

**Binding poses analysis**. The molecular docking of IacR with two IAA molecules was performed using UCSF Chimera v1.3 and the AutoDock Vina (http://vina.scripps.edu/). IacR and IAA charges were merged and the non-polar hydrogens and lone pairs were removed. In addition, hydrogens were added to the IacR structure. The exhaustiveness of the search was set to 8.

A total of 10 binding modes were generated and best two poses were selected based on the minimal energy scores shown in Supplementary Fig. 6. We provide further analysis of binding poses of IacR, MarR, and other phenolic compound sensing MarR proteins and demonstrated significant similarities in binding modes between these different MarR receptors involving first two alpha-helices that are in contact with ligands (Supplementary Fig. 6).

**Strains and plasmid constructions**. All plasmids in this study were created using isothermal Gibson assembly cloning. A middle-copy (~10–30 copies) episomal plasmid pGADT7 (Takara Bio Inc.) was used to increase the concentration of synthetic circuit components in order to buffer for the intrinsic molecular noise. This plasmid was further modified to accommodate three different auxotrophic selection markers (Leucine, Uracil, and Histidine; Supplementary Fig. 2a, b). GAL7 promoter and either standard CYC1 or ADH1 yeast terminators were cloned and combined with MarR modules in activator or repressor plasmids (Supplementary Fig. 2a, b). MarR and IacR were codon-optimized for yeast and synthesized using services delivered by Integrated DNA Technologies (IDT). The reporter plasmids include synthetic minimal promoters (synthesized with IDT) with previously identified MarR or IacR operator[25–28] sequences upstream TATA-box and minimal CYC1 promoter and fast-degradable UBG76V-EGFP (dEGFP)[31]. K1 toxin[47] amd Mig1 repression domain[48] were codon optimized and synthesized through IDT. K1 toxin replaced dEGFP gene on the reporter plasmid (Supplementary Fig. 2a, b). PCR reactions were performed using Q5 high fidelity polymerase (New England Biolabs). Correct PCR products were digested with DpnI (New England Biolabs) to remove the template and subsequently cleaned up with a DNA cleanup kit (Zymo Research) before Gibson assembly. Constructs were transformed in ultra-competent cells from *E. coli* DH5a strain using standard protocols. All plasmids were confirmed by colony PCR and validated with sequencing. The CEN.PK2-1C (a kind gift from Dr. Luis Rubio) yeast strain carrying integrated copy constitutively expressed mCherry reporter was used to prepare competent cells and transformation of plasmids using Frozen-EZ Yeast Transformation II Kit (Zymo Research). A single-copy genomic integration (Supplementary Fig. 10) was performed using EasyClone2.0 yeast toolkit[49] and insertion was confirmed by PCR and sequencing. Sequences of oligos and synthetic DNA used in this study are shown in Supplementary Table 1.

**Multiwell plate fluorescence measurements**. Overnight culture of the yeast in 2% sucrose low fluorescence media (Formedium, UK) was diluted 100 times and transfer to a 96-well plate with sucrose only or 2% sucrose +0.5% galactose conditions and a 2D gradient of SA and IAA concentrations. Plates were incubated at 30 °C overnight in the thermal incubator and well mixed by shaking before performing measurements. Co-culture experiments were performed as follows: sensitive strain and killer strain were grown separately overnight at 30 °C in 2% glucose media to stop K1 toxin production. Next day both culture were mixed in 1:1 ration in the induction media (2% sucrose + 0.5% galactose) and cultivated up to 24 h. Measurements were done with the Thermo Scientific™ Varioskan™ LUX multimode microplate reader after 24 h or were recorded every 10 min to generate time-lapse profile of dEGFP reporter. Optical density was set at an absorbance of 600 nm wavelength, the fluorescence excitation and emission light at 488 nm and 517 nm wavelength for dEGFP and 583 nm and 612 nm for Cherry, respectively.

**Time-lapse imaging, growth conditions, and data analysis**. Live-cell imaging was performed on the Automated inverted Leica DMi8 fluorescence microscope equipped with Hamamatsu Orca Flash V3 camera that was controlled by Micro-Manager v.2.0 (https://micro-manager.org/). Images were captured with 40x dry objective NA = 0.8 (Leica Inc.).Traps containing cells were imaged every 10 min on three different channels (DIC, GFP (Excitation: 488 Emission: 515, and mCherry Excitation: 583, Emission: 610) with CoolLed pE600 LED excitation source and standard Chroma epifluorescent filter set. Experiments were run for up to 72 h under the continuous supply of nutrients in the microfluidic device. Acquired images were initially processed in Fiji 2.0 (https://imagej.net/Fiji) using custom scripting to extract positions with exponentially growing yeast cells. Constitutively expressed mCherry marker was used to identify exponentially growing cells and used to derive normalized dEGFP fluorescence: dead or non-growing individuals were discarded by correcting dEGFP signal according to the formula $dEGFP/(dEGFP + mCherry)$. Each image was divided into 25 regions of interest (ROIs) and analyzed separately to isolate regions where cells were actively growing and could be tracked over time. The posterior analysis was done with custom R-studio scripts. Firstly, raw data were detrended using the detrend function from "pracma" R-studio v4.0.3 package and then smoothed with Savitzky–Golay Smoothing function (savgol), from the same package, with a filter length of 15 was applied and the signal was normalized between 0 and 1 to generate heatmaps across cell traps. Amplitudes were calculated with find peaks within the Process Data using the "findpeaks" function from "pracma" R package with nups and ndowns of 6, and periods were calculated by calculating distances between successive dEGFP peaks. Precision of response was formulated based on relative differences between peaks widths of input stimulus and dEGFP fluoprescence, respectively. For instance, the precision per $i$-th trap was calculated as follows: $precision^i = 1 - (width\_output^i - width\_input)/(width\_output^i + width\_input)$. Phase drift was calculated by comparing time differences of successive dEGFP peaks between cell communities in microfluidic device to derive inter-community synchronization measure. Cumulative autocorrelations traces and power spectrum densities were calculated on mean dEGFP trajectories per colony calculated for n-communities ($n > 20$, ~20,000 cells) using standard calculations with Matlab 2018b derived packages autocorr and Fast Fourier Transformation (fft).

**Reporting summary**. Further information on research design is available in the Nature Research Reporting Summary linked to this article.

## Data availability
All data that support the findings of this study are available in the manuscript and Supplementary Information. Raw Data for Figs. 1f, 3h, i, 4d–f, and 5g, h and Supplementary Figs. 2c, d, 4b, 8c, 9d, e, 10d, e, 11d, e, 13e, f, and 15c are provided as Source Data. Primer and plasmid part sequences are provided as Supplementary Table 1. Strains and plasmids are available from the corresponding author upon reasonable request. Source data are provided with this paper.

## Code availability
Image processing and analysis were performed using Fiji v2.0, Matlab R2018a, and R-studio v4.0.3. Subsequent analysis was performed with custom scripts that are available at the following link: https://github.com/PlantDynamicsLab/-Synchronization.git. Macromolecular dynamics was performed in OpenMM platform v7.4 and processing and visualization was done in VMD v1.9.3 and UCSF Chimera v1.3. Molecular docking was done with Autodock Vina, and structural alignment was performed with Molecular Evolutionary Genetics Analysis (MEGA) 11.

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

## Acknowledgements

We are grateful to L. Rubio for sharing yeast strain CEN.PK2-1C. Authors thank Mark Estelle for critical discussions at the early stage of the project. This work was supported by the Programa de Atraccion de Talento 2017 (Comunidad de Madrid, 2017-T1/BIO-5654 to K.W.), Severo Ochoa (SO) Programme for Centres of Excellence in R&D from the Agencia Estatal de Investigacion of Spain (grant SEV-2016-0672 (2017-2021) to K.W. via the CBGP). In the frame of SEV-2016-0672 funding M.A. received a PhD fellowship (SEV-2016-0672-18-3: PRE2018-084946) and O.P. is supported with a postdoctoral contract. K.W. was supported by Programa Estatal de Generacion del Conocimiento y Fortalecimiento Científico y Tecnologico del Sistema de I + D + I 2019 (PGC2018-093387-A-I00) from MICIU (to K.W.). UPM Plan Propio Predoctoral fellow finances M. G.N.

## Author contributions

S.P.G. and M.G.N. designed, performed most of experiments, and analyzed data. D.R.S., C.P.N., and O.P. contributed to plasmid and strain constructions, S.P.G. and M.A. performed multi-well plate fluorescence assays. M.G.N. performed microfluidics experiments. K.W. designed experiments analyzed data and supervised the project. All authors contribute to the manuscript writing.

## Competing interests

The authors declare no competing interests.
