## [Peer Review File · Nature Communications]

Reviewers' Comments:

Reviewer #1:

Remarks to the Author:

In the manuscript the authors develop a chemically-controlled hysteresis system to synchronise gene expression.

The system was probed in yeast and the components derive from multi antibiotic resistance (Mar) family.

the authors show the effective integration of multiple components in the system, and the achievement of synchronisation externally controlled by chemicals.

The conceived system is easy, the approach is clear and the implementation straightforward. The speculations about the functionality of the modules in the hysteresis system is supported by models.

The experimental setup based on microfluidics allow the realtime observation of the engineered system following the stimulations and the removal of the thereof.

The experiments support the hypothesis.

The overall manuscript reads quite well.

I would be curious to know from the authors how the system would translate in mammalian cells, and if a similar implementation is conceivable. The discussion should be improved with more attention on the avenues that this research opens up rather than summing the manuscript. They should include also a perspective in mammalian cells as they are more hard-to-engineer and robust systems.

I would also suggest to enhance the quality of the figures and to make the schematics bigger.

Reviewer #2:

Remarks to the Author:

This manuscript describes work that "synchronizes" gene expression in colonies of yeast. They do this by creating a two-input expression system and subjecting colonies to pulses of inducers.

Overall I think this study is a bit oversold. The authors use a lot of very technical terms to describe their system without much evidence to back their claims.

Specific comments:

1. I have a hard time describing the phenomenon shown here as synchronization - which in my mind is the emergent temporal coordination of dynamics due to internal processes (coupling). What's shown here is more like entrainment - the temporal coordination of dynamics via an external driving force. Even then, both terms, at least in dynamical systems theory, generally refer to oscillators (e.g. the entrainment of groups of circadian oscillators to the day/night cycle). I think at best what this phenomenon could be described as is a more robust temporal response to input.

2. With the above said, the question that should be raised now is, "Is a more uniform response to induction an interesting phenomenon." Perhaps, but it should probably be examined differently than what was done in this paper. For instance, the authors put the modified two-input expression system into cells and measured their dynamics (Fig. 1 g,h,i), and claim synchronization is due to the architecture. Well, they have not really proven that. They significantly altered one of the transcription factors (MarR). How does the now repressive MarR, by itself, behave? Perhaps it alone is the reason for the more robust response to input. This needs to be checked.

3. Figs. 1d and 1e really confuse me. Both SA and IAA should result in repression of the reporter in these experiments. I just can't tell when the ligands are present in the system. Was it just one pulse, or did the inducers get re-injected every 10 hours? Anyway I look at it, I just can't seem to see what the delay time is. For instance, if IAA were in the system during the 1st 5 hours of fig 1e, what the heck is going on with the response. It seems to repress ok, but then near hour 4 the system seems to predict the removal of IAA. These responses need to be greatly clarified.

4. It looks like the marR gene is mislabeled in Fig 1g.

5. I am not convinced that the data really support the finding of hysteresis. Generally, I think of hysteresis being the relaxing of a system onto one of multiple fixed points depending on the initial condition. In other words, the initial condition in relation to the basin boundaries determine which stable fixed point is reached. Granted, this may be more of a physics definition, but the definition that the past state affects the future trajectory just doesn't seem to cut it. If that were the case, any dynamical system that does not relax infinitely fast would meet that criterion. It's seems to me what is being observed here is more akin to a frequency dependent phase shift of a driven system.

6. The authors say that there is period doubling in the system, and hence surmise that the system is chaotic. This is quite a leap. Yes, successive period doubling bifurcations can lead to chaos, but I really see no evidence of that here. From the one plot (5c), it looks like a period was skipped - which is not terribly uncommon in noisy periodically driven systems. Further, if there really had been a period doubling bifurcation what was the parameter that changed? Why does the system not maintain that doubled period? Can the authors provide a dynamical systems model that predicts a period doubling bifurcation?

7. The authors state that their system is portable to other eukaryotic systems, but provide no evidence for this.

Reviewer #3:

Remarks to the Author:

In the manuscript "Synchronization of gene expression in eukaryotic communities through chemically-controlled hysteresis" the authors are exploring the process of synchronization by building a synthetic system that converts chemical signals into gene expression in yeast. They used two multi antibiotic resistance (Mar) receptors for indole-3-acetic acid (IAA) and salicylic acid (SA) and a fluorescent reporter and managed to obtain robust coordination of gene expression among yeast communities, which was proved robust also under challenging environmental conditions. Combining theory and experiments the authors analysed the dynamic hysteresis mechanism of the system, which contributed to provide precise spatial-temporal synchronization of gene expression patterns.

The study is nicely designed and accurately executed spanning several different computational and experimental domains. In particular, I will report here only on the computational part addressing the conformational switching of MarR and IacR using molecular simulations. This section of the study is well designed and instrumental to rationalise the origin of metastability associated with hysteresis in the systems. Using unbiased and metadynamics simulations starting from MarR high resolution structures the authors convincingly showed that the receptors can alternate between on and off states depending on the conditions, i.e., if DNA or small ligands are bound, respectively, while in apoform the receptors remain flexible, potentially able to adapt to the on or off state triggered by external interactions. I would not call this state "unstable" as it is not a single state but an ensemble of conformations. This ensemble would then rather represent the native unbound ensemble that has the potential to explore a much larger conformational space than the one

accessible in bound conditions.

I have two major comments regarding these results. Although they are sound, it is required that the authors provide a more solid proof of the reproducibility and convergence of their simulations results. For systems of this size, it is today very limited to run single simulations of few tens of nanoseconds. First, few replicas of the same simulation should be run to provide some statistical significance; second, longer simulations should be provided in the manuscript to ensure that proper convergence is reached. For example in Figure 2d, it is clear that some systems have not yet reached equilibrium as RMSD is still increasing. Few hundreds ns should be at least collected for each system, which is quite feasible for today's computational resources. My second major comment regards the set of simulations for IacR. While MarR simulations started from high-resolution structures, the authors built homology models of IacR based on MarR templates. They used an automated server to produce the models, but they need to mention and discuss the quality of these models as this will affect the following MD simulations. The authors need to report the sequence identity/similarity between IacR and MarR and discuss the regions that are modelled with higher or lower accuracy based on the questions asked in their study. Also the binding pose of IAA is predicted using a docking software, thus the authors need to discuss how reliable is this pose also in comparison with SA bound to MarR or any other experimental data available on IAA binding.

REVIEWER COMMENTS

Reviewer #1 (Remarks to the Author):

In the manuscript the authors develop a chemically-controlled hysteresis system to synchronize gene expression. The system was probed in yeast and the components derive from multi antibiotic resistance (Mar) family.

The authors show the effective integration of multiple components in the system, and the achievement of synchronization externally controlled by chemicals.

The conceived system is easy, the approach is clear and the implementation straightforward.

The speculations about the functionality of the modules in the hysteresis system is supported by models.

The experimental setup based on microfluidics allow the real-time observation of the engineered system following the stimulations and the removal of the thereof.

The experiments support the hypothesis. The overall manuscript reads quite well.

I would be curious to know from the authors how the system would translate in mammalian cells, and if a similar implementation is conceivable. The discussion should be improved with more attention on the avenues that this research opens up rather than summing the manuscript. They should include also a perspective in mammalian cells as they are more hard-to-engineer and robust systems.

I would also suggest to enhance the quality of the figures and to make the schematics bigger.

Author's response:

We thank Reviewer #1 for his positive assessment of the manuscript. As suggested, we added additional discussion on the use and portability of our compact system in animals and plants (Discussion, pages 8-9). These receptors can be tagged with any repression or activation domains from animals or plants (i.e. KRAB, Mxi, EAR, or similar motifs) to control the level of activation or repression. As these receptors bind directly to DNA, no other special components are required to translate chemical perception into transcriptional regulation. This is also the main reason why our system responds rapidly to chemical changes.

As opposed to prokaryotic systems, eukaryotes are hard to engineer concerning noise vs robustness tradeoff at the inter-population level. Therefore, we are convinced that this study will open new avenues to design and implement such robust systems in more complex eukaryotes.

Reviewer #2 (Remarks to the Author):

This manuscript describes work that "synchronizes" gene expression in colonies of yeast. They do this by creating a two-input expression system and subjecting colonies to pulses of inducers.

Overall I think this study is a bit oversold. The authors use a lot of very technical terms to describe their system without much evidence to back their claims.

Author's response:

We understand the concerns raised by Reviewer #2, and we do appreciate the commentaries presented below and we apologize for any confusions. However, as indicated by the other two Reviewers, we did provide a compelling evidence for the proposed mechanism using several independent experimental setups as well as theoretical modeling. To further strengthen our conclusions, we performed several additional experiments that we hope would address remaining concerns.

Specific comments:

1. I have a hard time describing the phenomenon shown here as synchronization - which in my mind is the emergent temporal coordination of dynamics due to internal processes (coupling). What's shown here is more like entrainment - the temporal coordination of dynamics via an external driving force. Even then, both terms, at least in dynamical systems theory, generally refer to oscillators (e.g. the entrainment of groups of circadian oscillators to the day/night cycle). I think at best what this phenomenon could be described as is a more robust temporal response to input.

Author's response:

We agree with Reviewer #2 that synchronization of dynamical systems (in physics) often refers to the synchronization of coupled oscillators. However, in nature, the synchronization may represent a general type of coordinated behavior found in collectives (i.e., bird flocks, fish schools, Wildebeest stampede, among others) and thus does not necessarily imply self-oscillating systems.

Concerning the entrainment, in this case we would typically talk about the entrainment of oscillators to an external signal. The idea is that individual oscillators can eventually lock with the peaks of applied stimulus. We apologize for any potential confusion and clarified the overall message in the revised version.

In our view, we present a system that is neither self-oscillatory nor entrained. In particular, we would like to stress that our system demonstrates the robust synchronization that occurs between cell populations (populations lock with each other) and not between population and external stimulus. Only the period of oscillations is dictated by external forcing which is somehow different from the synchronization of self-oscillatory cell systems to the driving signal.

Also, we would argue that this response is not only more robust to input, but it can rival with best-engineered systems in prokaryotes, and to our knowledge, such level of synchronization has never been achieved in engineered eukaryotes before.

Finally, knowing that transcriptional processes in eukaryotes are very noisy, the classical driven system would lock to different absolute phase delays and would neither maintain synchronized response for the prolonged time within a colony nor across different cell populations. This is clearly not the case in our system as responses remain very robust across populations over prolonged periods. Given that, we got naturally attracted by this phenomena and suggested the model that would allow for such robust entanglement.

2. With the above said, the question that should be raised now is, "Is a more uniform response to induction an interesting phenomenon." Perhaps, but it should probably be examined differently than what was done in this paper. For instance, the authors put the modified two-input expression system into cells and measured their dynamics (Fig. 1 g,h,i), and claim synchronization is due to the architecture. Well, they have not really proven that. They significantly altered one of the transcription factors (MarR). How does the now repressive MarR, by itself, behave? Perhaps it alone is the reason for the more robust response to input. This needs to be checked.

Author's response:

Again, we apologize for any confusion. We do not claim that synchronization is due to circuit architecture. It is most likely due to temporal rate-dependent hysteresis encoded in the hormone sensing mechanism which is not linked to the circuit architecture. Moreover, the modification of MarR was minor as we only exchanged the VP64 activation domain for an even smaller repression domain from Mig1 yeast repressor on the c-terminus of MarR. However, we performed a set of additional experiments that should clarify these concerns:

- 1) We integrated the system in the yeast genome and showed that the synchronization robustness does not depend on the copy number of MarRs (Supplementary Fig 10).
 - 2) We performed microfluidics experiments by fixing both or either of the chemicals and tested the synchronization robustness. Fixing both IAA and SA led to disorganized responses (Supplementary Fig 11a).
 - 3) Fixing SA at a low level and varying IAA led to weak synchronization whereas fixing IAA at low-level synchronization was lost. (Supplementary Fig 11b, c). Neither of these scenarios led to robust synchronization as opposed to the scenario in which both chemicals are dynamically changing (Supplementary Fig 11d,e). Therefore, neither MarR nor IacR alone are the reason for robust synchronization.
3. Figs. 1d and 1e really confuse me. Both SA and IAA should result in repression of the reporter in these experiments. I just can't tell when the ligands are present in the system. Was it just one pulse, or did the inducers get re-injected every 10 hours? Anyway I look at it, I just can't seem to see what the delay time is. For instance, if IAA were in the system during the 1st 5 hours of fig 1e, what the heck is going on with the response. It seems to repress ok, but then near hour 4 the system seems to predict the removal of IAA. These responses need to be greatly clarified.

Author's response:

We thank the reviewer for raising this issue and apologize for the confusion. We performed an additional experiment to gather more quantitative data from more traps on each of the Mar-type receptors tested. Also, we improved the visualization of chemical input in figures, and as requested, we indicated IAA or SA application in Fig 1d, e. Our results show that indeed the reporter response was delayed concerning input that is even more pronounced when we combined both receptors (Fig 1h). The response to chemical application is different from the response to chemical removal, thereby falling into the description of a hysteretic system.

4. It looks like the marR gene is mislabeled in Fig 1g.

Author's response:

As suggested this error has been corrected.

5. I am not convinced that the data really support the finding of hysteresis. Generally, I think of hysteresis being the relaxing of a system onto one of multiple fixed points depending on the initial condition. In other words, the initial condition in relation to the basin boundaries determine which stable fixed point is reached. Granted, this may be more of a physics definition, but the definition that the past state affects the future trajectory just doesn't seem to cut it. If that were the case, any dynamical system that does not relax infinitely fast would meet that criterion. It seems to me what is being observed here is more akin to a frequency dependent phase shift of a driven system.

Author's response:

We appreciate this comment and concur with the physics definition of hysteresis. We showed that response to increasing of input is different from the response to the input decrease (time lag or history dependence of the input) (Fig 1, Supplementary Fig 8), which is the common feature of hysteretic systems. However, unlike classical rate-independent hysteresis (memory), the type of hysteresis that we were able to record shows dynamic characteristics. In particular, it changes with the frequency of applied input. This type of transient hysteresis is called the rate-dependent hysteresis and has been described in piezoelectric actuator systems (Qin et al., 2017, Micromachines). We proposed a theoretical model with inclusion of rate-dependent hysteresis

and predicted the efficiency of such system for noise gating that complemented experimental observations.

It is also very unlikely that this system simply follow the frequency dependent phase shift since in the noisy environment, phase shifts would be eventually different for individual cells, and populations and thereby no such robust synchronization would be observed.

6. The authors say that there is period doubling in the system, and hence surmise that the system is chaotic. This is quite a leap. Yes, successive period doubling bifurcations can lead to chaos, but I really see no evidence of that here. From the one plot (5c), it looks like a period was skipped - which is not terribly uncommon in noisy periodically driven systems. Further, if there really had been a period doubling bifurcation what was the parameter that changed? Why does the system not maintain that doubled period? Can the authors provide a dynamical systems model that predicts a period doubling bifurcation?

Author's response:

We apologize for making an unintentional leap. We revised the text to include speculation about the possibility for the chaos that would result from successive period doublings. Yet, we do not observe period doublings in other experiments neither in an additional control experiment in which sensitive strain was coculture with a strain carrying empty plasmids (Fig 5 and Supplementary Fig 15). Therefore, the two-period doublings are specific to cocultures with toxin-producing strains. The route to chaos requires successive period doublings contrary to the maintenance of a single doubling event. This is precisely what we observed in our experiments with toxin-producing strain. A further independent investigation would help us to identify mechanisms that would potentially lead to those doublings but is out of scope of this work. We are currently performing independent study to test different toxins and their effect on population dynamics.

We would propose a simple model based on a logistic map of growth with the external periodic stimulus(chemical):

$$\text{Stimulus}(t) = 1 + \text{square_wave_generator}(\text{frequency}=2*\text{PI}, \text{duty}=50\%);$$

$$X(t+1) = (2 + \text{Stimulus}(t))*X(t)*(1-X(t));$$

where X is a population of GFP/toxin producing cells in a given time point t.

This simple model can predict series of period doublings that is dependent on the chemical stimulus. However, we would prefer to not add this model in the current manuscript as we work on independent story in which we focus on effect of different toxins on population dynamics.

7. The authors state that their system is portable to other eukaryotic systems, but provide no evidence for this.

Author's response:

Although we did not yet tested our system in other eukaryotes, we are quite confident there will be no major issue by using the same architecture in other eukaryotes such as animals and plants. We improved discussion on that matter in the revised manuscript (Discussion, pages 8 and 9). For instance, VP64 trans-activation domain that we used in our study is equally functional in most eukaryotes. Concerning repression domains, other domains such as KRAB, EAR, and active units of histone deacetylase (i.e. TPL/TUP1/GROUCHO) could be attached on the c-terminus of MarRs, In fact, we are currently investigating various chromatin modulators from fungi, plants, and animals that were attached to MarRs, and we have obtained promising preliminary results. Other than this our system does not require any additional components. To further test the universality of our system we collaborate with plant scientists to test these receptor systems in plants.

Reviewer #3 (Remarks to the Author):

In the manuscript “Synchronization of gene expression in eukaryotic communities through chemically-controlled hysteresis” the authors are exploring the process of synchronization by building a synthetic system that converts chemical signals into gene expression in yeast. They used two multi antibiotic resistance (Mar) receptors for indole-3-acetic acid (IAA) and salicylic acid (SA) and a fluorescent reporter and managed to obtain robust coordination of gene expression among yeast communities, which was proved robust also under challenging environmental conditions. Combining theory and experiments the authors analysed the dynamic hysteresis mechanism of the system, which contributed to provide precise spatial-temporal synchronization of gene expression patterns.

The study is nicely designed and accurately executed spanning several different computational and experimental domains. In particular, I will report here only on the computational part addressing the conformational switching of MarR and IacR using molecular simulations. This section of the study is well designed and instrumental to rationalise the origin of metastability associated with hysteresis in the systems. Using unbiased and metadynamics simulations starting from MarR high resolution structures the authors convincingly showed that the receptors can alternate between on and off states depending on the conditions, i.e., if DNA or small ligands are bound, respectively, while in apoforn the receptors remain flexible, potentially able to adapt to the on or off state triggered by external interactions. I would not call this state “unstable” as it is not a single state but an ensemble of conformations.

This ensemble would then rather represent the native unbound ensemble that has the potential to explore a much larger conformational space than the one accessible in bound conditions.

Author's response:

We thank Reviewer #3 for his positive assessment of the manuscript. We agree with these suggestions and revised this definition. However, we believe that “unstable” (in classical definition) reflects that there is no permanent state as the system oscillates between open and closed conformations, which can be seen in three independent MD replicates (Fig 2d).

I have two major comments regarding these results. Although they are sound, it is required that the authors provide a more solid proof of the reproducibility and convergence of their

simulations results. For systems of this size, it is today very limited to run single simulations of few tens of nanoseconds. First, few replicas of the same simulation should be run to provide some statistical significance; second, longer simulations should be provided in the manuscript to ensure that proper convergence is reached. For example in Figure 2d, it is clear that some systems have not yet reached equilibrium as RMSD is still increasing. Few hundreds ns should be at least collected for each system, which is quite feasible for today's computational resources.

Author's response:

As suggested, we performed MD simulations for a longer time (300ns) in three replicates for each system which was quite challenging due to our limited computational resources. In the revised version of the manuscript, we provided updated figures (Fig 2 and Supplementary Fig 7) and main text (page 5; in yellow). The consensus is that our observations were further confirmed in three independent replicas.

My second major comment regards the set of simulations for IacR. While MarR simulations started from high-resolution structures, the authors built homology models of IacR based on MarR templates. They used an automated server to produce the models, but they need to mention and discuss the quality of these models as this will affect the following MD simulations. The authors need to report the sequence identity/similarity between IacR and MarR and discuss the regions that are modelled with higher or lower accuracy based on the questions asked in their study. Also the binding pose of IAA is predicted using a docking software, thus the authors need to discuss how reliable is this pose also in comparison with SA bound to MarR or any other experimental data available on IAA binding.

Author's response:

We thank Reviewer #3 for raising this issue. Now we provide the analysis of IacR structure prediction using several independent scoring metrics (Supplementary Fig 5). The predictions in regions involved in DNA binding and dimerization are of high confidence (negative QMEAN scores), and generally, the structure prediction fidelity closely matches native folds (Z -score < 2). However, this has to be confirmed by X-ray crystallization studies which we intend to perform in the future.

Now the revised version contains the information about free binding energies for native SA pose with MarR and as well as predicted IAA pose in IacR (Supplementary Fig 6). We found that the IAA pose is very similar to SA poses in other Mar-type receptors with resolved crystal structures (Supplementary Fig 6) and involves contacts with the first two alpha-helices of each monomer. We discuss those results on page 5 of the revised manuscript and Supplementary Figs 5 and 6 legends.

Reviewers' Comments:

Reviewer #2:

Remarks to the Author:

The authors have addressed most of my issues. I still think some of the claims are somewhat tenuous, especially the period doubling, but I am not going to press the issue.

Reviewer #3:

Remarks to the Author:

The authors well responded to my major concerns and now the molecular modelling and simulation parts of the work are very robust. One final remark: to ensure reproducibility the authors should report in the method section the cutoff used for treating the non-bonded interactions; the commonly used value is 12 Å.

REVIEWERS' COMMENTS

Reviewer #2 (Remarks to the Author):

The authors have addressed most of my issues. I still think some of the claims are somewhat tenuous, especially the period doubling, but I am not going to press the issue.

Author response: We thank Reviewer for his valuable comments that helped us to improve the manuscript.

Reviewer #3 (Remarks to the Author):

The authors well responded to my major concerns and now the molecular modelling and simulation parts of the work are very robust. One final remark: to ensure reproducibility the authors should report in the method section the cutoff used for treating the non-bonded interactions; the commonly used value is 12 Å.

Author response: Indeed this is the standard value that we used and this information is now included in the method's section.